# A Case of Trauma-Induced *Falciformispora lignatilis* Eumycetoma in a Renal Transplant Recipient

**DOI:** 10.3390/tropicalmed6030144

**Published:** 2021-08-03

**Authors:** Maxwell Olenski, Catriona Halliday, James Gullifer, Elena Martinez, Amy Crowe, Harsha Sheorey, Jonathan Darby

**Affiliations:** 1Infectious Diseases Department, St Vincent’s Hospital, Melbourne, VIC 3065, Australia; amy.crowe@svha.org.au (A.C.); jonathan.darby@svha.org.au (J.D.); 2Microbiology Department, St Vincent’s Hospital, Melbourne, VIC 3065, Australia; harsha.sheorey@svha.org.au; 3Institute of Clinical Pathology and Medical Research, NSW Health Pathology, Westmead, NSW 2145, Australia; Catriona.Halliday@health.nsw.gov.au (C.H.); elena.martinez@health.nsw.gov.au (E.M.); 4Pathology Department, St Vincent’s Hospital, Melbourne, VIC 3065, Australia; james.gullifer@svha.org.au

**Keywords:** mycology, mycetoma, eumycetoma, transplant, immunity

## Abstract

Mycetoma is a chronic, granulomatous, subcutaneous infection caused by several species of fungi and soil-inhabiting bacteria, and is divided into eumycetoma and actinomycetoma, respectively. Endemicity is described with worldwide distribution within the “mycetoma belt”; however, the global burden is ill-defined. Mycetoma is rare in Australia, with only a few published case reports. Over time, the breadth of eumycetoma pathogens has expanded with local epidemiology accounting for variations in regional prevalence. Direct inoculation of pathogens typically heralds the triad of subcutaneous mass, sinus formation and discharging grains. We describe a case of eumycetoma in a 48-year-old male Filipino renal transplant recipient who presented with a painless slow-growing elbow lesion. Ultrasonography revealed two ovoid masses and surgical excision ensued. Histopathology revealed necrotising granulomata with numerous chestnut-brown thick-walled cells, septate hyphae, and occasional grains. On suspicion of localised chromoblastomycosis, the isolate was sent to a reference laboratory which identified the fungus as *Falciformispora lignatilis*, an organism not hitherto associated with human infection. Amongst the solid organ transplant cohort, similar atypical presentations have been described. Clinicians need to consider eumycetoma where an epidemiological link with the tropics exists, especially in atypical presentations in transplant recipients, including absent preceding trauma.

## 1. Introduction

Mycetoma is a chronic, granulomatous, subcutaneous infection caused by several species of fungi and soil-inhabiting bacteria, and is divided into eumycetoma and actinomycetoma, respectively. It is endemic within the “mycetoma belt”—ranging from 30° N and 15° S—however, the global burden is ill-defined, and mycetoma was recognised as a neglected tropical disease by the World Health Organisation in 2016 [1,2]. First described in India, other high incidence countries include Sudan, Somalia, and Mexico [3]. Direct inoculation of causative pathogens typically heralds grain formation within the skin and soft tissues, with progression to subcutaneous nodules, masses and suppuration via sinus tracts [3]. Although mycetoma typically affects the lower limbs, lesions can develop at other sites including the arms, hands and torso.

Mycetoma is rare in Australia, with only a few published case reports [4]. We describe a novel pathogen in a case of eumycetoma in a renal transplant recipient.

## 2. Case Report

A 48-year-old male was admitted to a tertiary referral centre for elective excision of a right-sided elbow lesion that had developed over four months.

The patient was known to the Nephrology Unit following a renal transplant in 2017 for end-stage kidney disease in the context of primary focal segmental glomerulosclerosis. The initial period of post-transplant immunosuppression was complicated by pulmonary aspergillosis, for which he completed six months of voriconazole therapy with significant interval clinical and radiological improvement as of November 2018.

Other comorbidities included well-controlled hypertension, dyslipidaemia, steroid-induced osteoporosis and gastro-oesophageal reflux. Immunosuppressive therapy included 5 mg of prednisolone daily, everolimus 0.5 mg twice daily, and 1 mg extended-release tacrolimus daily.

The patient was born in the Philippines prior to migrating to Australia at age 36, with biannual return trips to visit friends and family. He was neither a smoker nor drank alcohol to excess, and was previously employed in maintenance. He reported no exposure to mangroves or fishing, and did not spend significant time gardening.

The patient initially noticed a painless soft tissue mass on the lateral aspect of his right elbow following minor blunt trauma by a doorknob sustained in Australia in 2019. There was no skin breech, and the lesion grew in size over the ensuing months. Throughout this period he remained systemically well, with no other discernible soft tissue masses.

Baseline blood examination was unremarkable, with stable renal function (Creatinine 169 µmol/L [reference range 64–104 µmol/L], eGFR 41 mL/min/1.73 m^2^ (reference range > 90 mL/min/1.73 m^2^)) and normal inflammatory markers, including white cell indices and a C-reactive protein < 5 mg/L (reference range < 5 mg/L).

Ultrasonography revealed two small circumscribed ovoid masses, suggestive of reactive lymphadenopathy (Figure 1A). Following surgical review, a 1.5 cm mobile mass was removed which was felt to macroscopically to represent reactive lymphadenopathy. The tissue was set up for routine, mycobacterial and fungal culture, and sent for histopathological examination. No organisms were seen nor cultured from Gram and Zeihl-Neelson stains; however, filamentous fungi were identified on Blankophor preparation.

Histopathological examination revealed extensive necrotising granulomatous reaction with a mixed acute and chronic inflammatory cell population, including neutrophils and multi-nucleated giant cells. The granulomata contained numerous chestnut-brown thick-walled cells, consistent with Medlar bodies, as well as septate hyphae, and occasional grain formation. Periodic acid-Schiff (PAS) staining showed these elements to be PAS-positive, consistent with fungal origin (Figure 1B,C).

On suspicion that the lesion represented a localised chromoblastomycosis, the specimen was sent to the Mycology Laboratory at the Centre for Infectious Diseases and Microbiology Laboratory Services (ICPMR–NSW Health Pathology, Westmead Hospital, NSW) for culture and identification. After 10 days, scant growth was observed on Sabouraud’s dextrose agar containing antibiotics at 30 °C, and attempts at inducing sporulation proved fruitless with nondescript fungal hyphae seen on lactophenol cotton blue staining. After a further 4 days, velvety colony growth was observed, which was radially folded, initially marble white and becoming pale grey to purple, with brown exudate produced on the colony surface and dark brown reverse (Figure 1D).

For species identification, the internal transcribed spacer (ITS) 1, 5.8*S* and ITS2 regions, and the D1/D2 region of the 28*S* (large subunit) rDNA gene were amplified by PCR using previously published primers and protocols (CLSI MM18). Using the BLASTn search within the Westerdijk Fungal Biodiversity Institute Database (https://wi.knaw.nl/page/Pairwise_alignment, accessed on 20 May 2020) the sequences of the resulting ITS amplicon (GenBank MZ268113) showed 99.4% identity (529/532 base pairs) to *Falciformispora lignatilis* (KF432942). The next closest match was to *Falciformispora tompkinsii* (KF015670) with 96.5% identity (464/481 base pairs). Although the D1/D2 regions are more conserved, the sequencing results (542 base pairs, GenBank MZ268119) supported the identification of *F. lignatilis* with 100% identity to GU371826.1 and 99% identity (539/542) to *F. tompkinsii* (MH8792969.1). Partial 28*S* and ITS sequences for reference *Falciformispora* isolates and the clinical isolate were concatenated, aligned and a Neighbour-joining tree were constructed using Geneious Prime (version 11.0.3) (Figure 2).

Antifungal susceptibility testing was performed using the Sensititre^TM^ YeastOne^TM^ YO10 plate (ThermoFisher Scientific) which is based on Clinical and Laboratory Standards Institute (CLSI) methodology, with numerous studies showing reproducibility and good correlation between Sensititre^TM^ and CLSI for non-Aspergillus moulds [5]. Unfortunately, no MICs were able to be generated due to poor growth in the Sensititre^TM^ YeastOne^TM^ broth.

The patient was commenced on 300 mg of delayed-release oral posaconazole daily for an intended 6-month course, with monitoring of both his posaconazole and tacrolimus levels throughout the duration of therapy.

## 3. Discussion

Eumycetoma tends to affect people living in tropical and subtropical regions, though sporadic cases have been reported worldwide including in temperate regions. As such, risk factors for acquisition include environmental exposure to causative agents, inherited predisposition to infections, and immunosuppression.

Over time, the breadth of pathogens associated with eumycetoma has continued to widen [2]. At least 40 dematiaceous fungi causing eumycetoma have been described and are divided based on the pigment of grains they produce (Table 1). Local epidemiology, including climate, vegetation, rainfall and soil type account for variations in prevalence across regions.

All fungi known to cause eumycetoma belong to the phylum Ascomycota. Amongst the order Pleosporales, based on multi-gene phylogeny, *Leptosphaeria senegalensis* and *Leptosphaeria tompkinsii* were found to cluster with the Trematosphaeriaceae—with *Falciformispora lignatilis* the closest relative—and have thus been designated *Falciformispora senegalensis* and *Falciformispora tompkinsii*, respectively [8]. *F. lignatilis* has thus far only been recovered from mangrove wood, terrestrial oil palm and fresh water decorticated woody debris in North America and Thailand [8,9], and, to the best of our knowledge, has not hitherto resulted in human infection.

Traumatic inoculation of eumycetoma pathogens from agricultural exposure typically begets infection, including thorns, plant materials and watery habitats, especially amongst farmers and fishermen. Absence of a known predisposing injury is well described, and even innocuous injury without penetrating trauma may render body regions vulnerable (viz. locus minoris resistentiae). A classic triad of soft tissue mass, sinus tracts and macroscopic grains is useful for establishing a clinical diagnosis. Lesions tend to evolve over months to years, and most frequently involve the feet (68.7%), leg (9.9%), trunk (6.1%) and arm (4.0%) [3].

Laboratory diagnosis includes standard fungal culture of grains and sinus fluid to identify associated microbial pathogens [7]. In the absence of sinus tracts or grains, tissue biopsy is needed to evaluate for presence of intralesional grains. Complementary molecular-based diagnostic techniques utilise sequencing of highly conserved regions within the ITS region of causative agents of eumycetoma, and are amplified using panfungal primers. This approach allows comparison with findings catalogues within Genbank, and provides insight into the biology and pathogenicity of such fungi. Indeed, such techniques have allowed for identification of previously underspeciated eumycetoma pathogens [10]. In general, however, such modalities are not available in areas of endemicity.

In addition, three histopathological types of tissue reactions are recognised, which relate the presence or absence of grains to acuity of the inflammatory infiltrate, running the gamut from neutrophilic to granulomatous. Regional imaging to define the extent of disease, including the presence of bony involvement, is also useful.

The current standard of care is a combined medical and surgical approach [11]. Limited antifungal susceptibility data are available for most fungi causing eumycotic mycetoma. Historically, ketoconazole and itraconazole formed the mainstay of eumycetoma antifungal therapy, though in vitro susceptibility and in vivo clinical response is variable. In general, novel triazoles have favourable in vitro and in vivo efficacy, though the chasm between in vitro data and clinical response remains agape [12]. Nonetheless, whilst antifungal susceptibilities of different azoles vary amongst aetiological agents, posaconazole appears to have the lowest MIC (range 0.01–1 μg/mL amongst coelomycetes) [2,6]. The optimal duration of antifungal therapy is currently unknown, though emerging data suggest utility in MRI, PET/CT and ß-D-glucan to monitor for extension and relapse [13].

A number of cases of eumycetoma in solid organ transplant recipients have been described. Meis et al. report a case of *Madurella mycetomatis* in a renal transplant recipient who migrated to Holland from Curaçao 40 years prior to presentation, whereby he presented with atraumatic foot swelling without sinus tract nor grain formation, diagnosed following surgical excision [14]. Similarly, in 2015 McGrogan et al. present another atypical case of eumycetoma in a Ghanaian man in the absence of trauma, as well as summarise the characteristics of transplant-associated mycetomas [15]. Together with our case, these bolster the need to consider eumycetoma where an epidemiological link with the tropics exists, however remote, as well highlight the oft atypical nature of presentations amongst a transplant cohort, including an absent clinical triad and memory of preceding trauma.

The differential diagnosis of eumycetoma includes a range of pathologies, including foreign body granuloma, cystic lesions and soft tissue neoplasm. In addition, in the absence of sinus tract formation, and in the appropriate clinical and epidemiological setting, sporotrichosis, chromoblastomycosis, cutaneous leishmaniasis and tuberculosis need to be considered [16]. Knowledge of key differentiating features, including epidemiology, mode of acquisition and clinical and histological features is paramount.

## 4. Conclusions

Mycetoma is an important subcutaneous infection in tropical and subtropical countries, often characterised by a clinical triad of subcutaneous mass, sinus formation, and discharging grains. Eumycetoma pathogens are well-described and tend to follow a more indolent course compared to their bacterial counterparts.

We propose *Falciforma lignatilis* be included as a novel agent of eumycetoma, as well as highlight the atypical nature of presentations amongst solid organ transplant recipients.

Clinicians should be aware of the possibility of mycetoma in the immunosuppressed post-transplant patient who frequents areas of endemicity. Whilst the optimal choice of antifungal agents is currently not known, newer agents such as voriconazole and posaconazole show promise.

## Figures and Tables

**Figure 1 tropicalmed-06-00144-f001:**
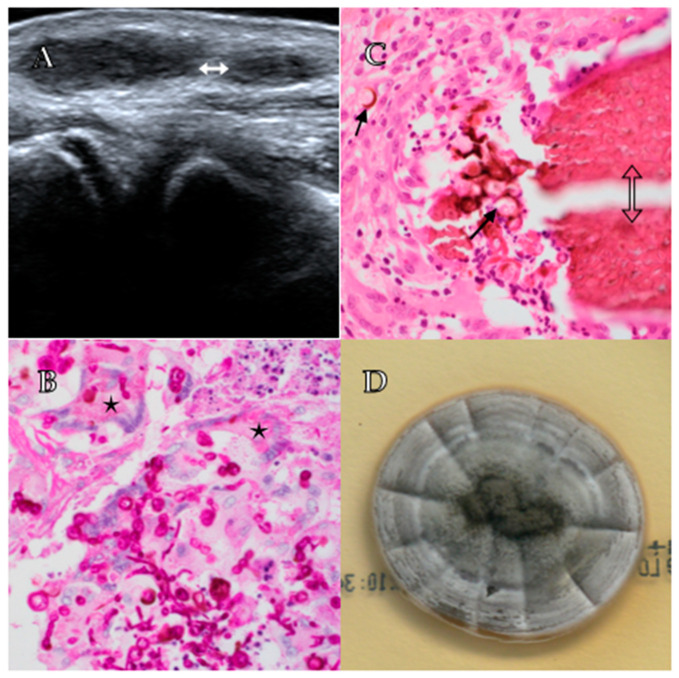
Panel (**A**) shows a right elbow ultrasound demonstrating two small ovoid lesions (white double-headed arrow). Panel (**B**) shows septate hyphae and multinucleated giant cells containing fungal elements (stars) at 600× power on PAS staining. Panel (**C**) shows a grain (hollow double-headed arrow) with surrounding thick walled, brown fungal cells (arrows) at 600× power on H&E staining. Panel (**D**) shows fungal growth after 14 days on Sabouraud’s dextrose agar demonstrating grey and purple colonies with radial folding.

**Figure 2 tropicalmed-06-00144-f002:**
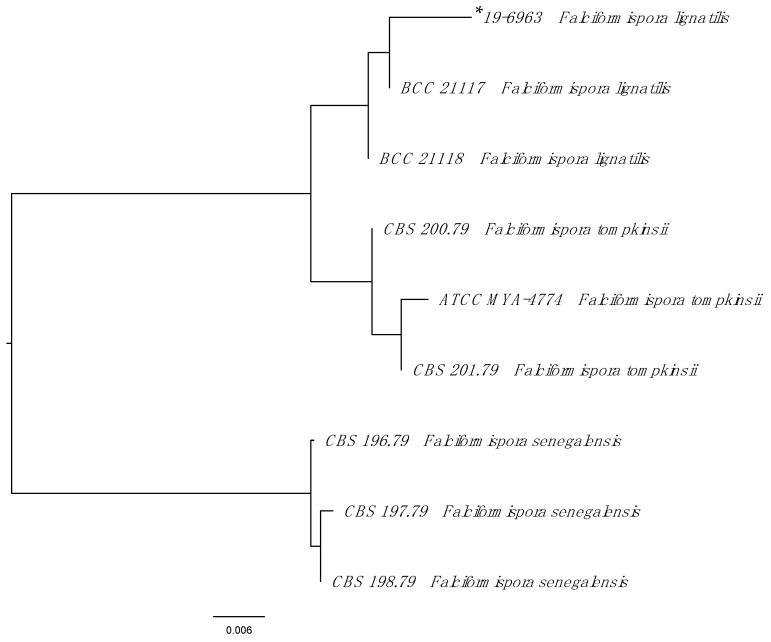
Phylogenetic tree of *Falciformispora* species reported to cause eumycetoma. The Neighbour-joining tree was constructed from combined partial 28*S* and internal transcribed spacer sequences. See Ahmed et al. [6] for a more comprehensive phylogenetic overview of all agents of eumycetoma. * Indicates clinical strain.

**Table 1 tropicalmed-06-00144-t001:** Causative agents of eumycetoma.

Grain Colour	Causative Organism
Black	*Madurella* spp. (*M. mycetomatis*, *M. fahali*, *M. pseudomycetomatis*, *M. tropicana*)
	*Falciformispora* (formerly *Lepstosphaeria*) spp. (*F. senegalensis*, *F. tompkinsi*, *F. lignatilis*)
	*Curvuralia* spp.
	*Exophiala* spp.
	*Phaeoacremonium* spp.
	*Phialophora verrucosa*
	*Biatriospora mackinnonii* (formerly *Pyrenochaeta mackinnonii*)
	*Trematosphaeria grisea*
	*Medicopsis romeroi* (formerly *Pyrenochaeta romeroi*)
Pale/White/Yellow	*Scedosporium apiospermum* (formerly *Pseudallescheria apiosperma*)
	*Acremonium* spp.
	*Aspergillus* spp.
	*Fusarium* spp.

Modified from Ahmed et al. [7].

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
