# Peer review of "A Case of Trauma-Induced Falciformispora lignatilis Eumycetoma in a Renal Transplant Recipient"

_tropicalmed, 2021, doi:10.3390/tropicalmed6030144_

Round 1

Reviewer 1 Report

The manuscript by Olenski et al describes a mycetoma case caused by a new fungus. Since there is a paucity of information about this neglected disease, this is an important report that can help other clinicians and mycologists in the future.

A few things are necessary to improve the quality of the manuscript:

1) Title: There is a typographic error in the fungus name.

2) Figure 1: It should be interesting to include a panel showing the microscopic features (from a slide culture) of the fungus.

3) The antifungal susceptibility testing must be done using the CLSI or EUCAST methods. The Sensititre Yeast One is not suitable for a filamentous fungus.

4) Discussion: Authors should include comments about the case described within the entire discussion. For instance: authors say: "A classic triad of soft tissue mass, sinus tracts and macroscopic grains is useful for establishing a clinical diagnosis." Were any of these observed in the patient described? This should be done in the entire discussion, because in the present form, the discussion looks like a review of the literature.

Reviewer 2 Report

From only few years, the WHO recognizes mycetoma as a neglected tropical disease of interest.

Cases with well-documented molecular identification as this report contribute to increase epidemiological knowledge about this disease.

Imported cases in transplant recipients that are native from endemic areas or travel in there may occur. Thus, such cases are of interest.

The authors report the first case of Falciformispora lignitalis eumycetoma in a renal transplant recipient. The histopathology and mycological identification, including molecular, are well documented. However, some additional information should improve the case report.

The nature (adenopathy or hypodermal tissue…) of the 1.5 cm removed mass (lines 67 to 72 and figure 1) is not obvious. This could be specified, at least in the Figure 1 legend.

For Figure 1, please indicate the ovoid lesions with a double-headed arrow and size if available.

As the grain is the histological signature of mycetoma it should be pointed by an arrow or an arrow head.

By the way, speaking about Medlar bodies (legend of figure 1 and text, line 82) might be confusing as they are usually considered as pathognomonic of chromycosis. I suggest replacing “Medlar bodies” with a short description of the fungal elements as “thick walled, brown fungal cells without septa”.

In Figure 2, I suppose that the strain from this report is 19-6963. A star, pointing it, would be appreciate.

Have the authors deposited the sequence of this strain in a database? If not, this should be done and the reference (e.g. GenBank NCBI number) should be specified.

As a risk of extension or relapse exists, data about anatomic extension and radiological and biological monitoring are important. Did the authors performed a BD glucane dosage (high levels may be seen in eumycetomas)? Maybe a serum is still available. If positive, it would be a simple parameter to use for monitoring. To asses anatomical extension MRI (there are specific features for eumycetoma) or, better PET CT, are more accurate than ultrasonography. Did the authors planed any?

The patient was treated with posaconazole. Form (tablet or suspension) and posology should be specified as well as the dosing values if available.

Table 1. Fusarium spp. is also a causative organism of eumycetoma (as frequent as Acremonium spp.); it should be included in the table.

As the molecular identification is a strong point of the case report, it deserves to be highlighted in the discussion paragraph.

Line 169 I would say “absence of memory of preceding trauma” instead absence of preceding trauma”

Is there any supplementary material? The link (line 188) does not work
